# Age-structured non-pharmaceutical interventions for optimal control of COVID-19 epidemic

Quentin Richard[1], Samuel Alizon[1], Marc Choisy[1,2,3], Mircea T. Sofonea[1], Ramsès Djidjou-Demasse[1] *

1 MIVEGEC, Univ. Montpellier, IRD, CNRS, Montpellier, France, 2 Centre for Tropical Medicine and Global Health, Nuffield Department of Medicine, University of Oxford, United Kingdom, 3 Oxford University Clinical Research Unit, Ho Chi Minh, Vietnam

* ramses.djidjoudemasse@.ird.fr

**Data Availability Statement:** All relevant data are within the paper and its Supporting information files. The code (with the Julia Programming Language) used to simulate the model can be

## Abstract

In an epidemic, individuals can widely differ in the way they spread the infection depending on their age or on the number of days they have been infected for. In the absence of pharmaceutical interventions such as a vaccine or treatment, non-pharmaceutical interventions (*e.g.* physical or social distancing) are essential to mitigate the pandemic. We develop an original approach to identify the optimal age-stratified control strategy to implement as a function of the time since the onset of the epidemic. This is based on a model with a double continuous structure in terms of host age and time since infection. By applying optimal control theory to this model, we identify a solution that minimizes deaths and costs associated with the implementation of the control strategy itself. We also implement this strategy for three countries with contrasted age distributions (Burkina-Faso, France, and Vietnam). Overall, the optimal strategy varies throughout the epidemic, with a more intense control early on, and depending on host age, with a stronger control for the older population, except in the scenario where the cost associated with the control is low. In the latter scenario, we find strong differences across countries because the control extends to the younger population for France and Vietnam 2 to 3 months after the onset of the epidemic, but not for Burkina Faso. Finally, we show that the optimal control strategy strongly outperforms a constant uniform control exerted over the whole population or over its younger fraction. This improved understanding of the effect of age-based control interventions opens new perspectives for the field, especially for age-based contact tracing.

## Author summary

COVID-19 infected individuals differ in the way they spread the infection depending on their age or on the number of days elapsed since the contamination. This individual heterogeneity can impact the design of public health control measures to contain epidemics. Using optimal control theory, we identify a strategy that minimizes deaths and costs due to the implementation of the control measures themselves. We also implement this

**Funding:** RDD, SA, MTS was partly supported by the Occitanie region and the ANR (https://anr.fr/en/) grant PHYEPI. QR received support from the ANR STORM (grant agreement 16-CE35-0007). The funders had no role in study design, data collection and analysis, decision to publish, or preparation of the manuscript.

**Competing interests:** The authors have declared that no competing interests exist.

strategy for three countries with contrasted age distributions (Burkina-Faso, France, and Vietnam). This strategy consists in rapidly intervening in older populations to protect the older people during the initial phase of the epidemic and (if the cost is intermediate or low) to control the epidemic, before progressively alleviating this control. Interventions in the younger population can occur later if the cost associated with the intervention is low. Such interventions targeted at younger people aim at suppressing the epidemic.

## Introduction

Following its emergence in December 2019, COVID-19 has become an international public health emergency [1]. The infection has many similarities with that caused by influenza virus regarding its clinical manifestations and transmission mechanisms [1]. However, contrary to seasonal influenza, COVID-19 has become pandemic by spreading rapidly among completely naive host populations, *i.e.* with no pre-existing immunity [2–5]. Initially, no pharmaceutical interventions such as vaccines or treatments were available and it usually takes several months before their deployment. For this reason, non-pharmaceutical intervention (NPI) strategies, such as social distancing, are key to controlling the pandemic [6].

When an intervention is summarized by one or few parameter values, identifying an optimal strategy according to some criterion variable can readily be done, *e.g.* using a gradient approach [7]. Things become more challenging when the intervention parameter value is a function of time. Optimal control theory [8], specifically addresses this issue by identifying a function of time such that some criterion is optimized. This has allowed studies to identify optimal non-pharmaceutical interventions to control infectious diseases such as influenza and COVID-19 [9–12]. However, a strong limitation of these studies is that they all ignore at least one aspect of the host population structure. First, infection parameters (contagiosity, recovery) vary with infection age, *i.e.* depending on the number of days since infection. Second, hosts vary in age. The latter point is particularly important because, in addition to being a function of time since the onset of the outbreak, optimal strategies involving physical distancing can also vary depending on host age [13–17]. Accounting for two dimensions, time and host age, make the optimization procedure much more challenging because optimal control theory is usually applied to ordinary differential equations (ODEs) –something very common– while here we are working on partial differential equations (PDEs) –which is less common, and much more challenging. Here, we address this challenge and identify interventions varying in intensity with time and host age that significantly reduce morbidity associated with COVID-19 at a minimal cost. Furthermore, we compare the situation in countries with contrasted age-structure, namely Burkina-Faso, France, and Vietnam, to show how this affects optimal strategies.

The age structure of the population is a known key determinant of acute respiratory diseases, especially when it comes to infection severity. For example, children are considered to be responsible for most of the transmission of influenza virus [18] but the related hospitalization and mortality burden is largely carried by people of ages over 65 years [19, 20]. While much remains unknown about the COVID-19 epidemics, evidence to date suggests that mortality among people who have been tested positive for the coronavirus is substantially higher at older ages and near zero for young children [3, 21]. Moreover, the infectiousness of an individual has been reported to vary as a function of time since infection [22], which is known to affect epidemic spread [23–26].

Our model for the disease stage-progression is structured both by a (continuous) age of the host and a (continuous) age of infection. A variety of epidemiological models allow for one or the other type of structure [27–30], starting with a seminal article from the 1920s [23]. However, models allowing for a double continuous structure are rare [30–37], even though it is particularly suited to investigate infections such as COVID-19, with strong effects of host and infection age. Indeed, in addition to taking into account the age structure of the host population, as well as the gradient of disease severity from mild to critical symptoms, the model readily captures the variation in infectiousness as a function of the time since infection. From a theoretical point of view, age-structured models have been proposed to investigate the spread of acute respiratory diseases [38–42], and some rare models of acute respiratory diseases consider both structures as continuous variables [30, 32], although not in the context of optimal control theory.

We first introduce the mathematical model and define its parameters and outputs. Next, we characterize the optimal control strategy that minimizes the number of deaths as well as the cost due to the implementation of the control strategy itself. The main body of the results then follows. We first analyze the epidemic spread without any intervention for three countries with contrasted age distributions (Burkina-Faso, France, and Vietnam). Second, we compare the performance of optimal control in terms of deaths and hospitalizations for different costs of the control measure. Finally, we compare our optimal control strategy to two other strategies that use the same amount of resources to control the outbreak. Finally, we discuss our model in results in the context of the current pandemic and identify perspective for future work.

## Materials and methods

### An age-structured epidemiological model

**Model overview.**    We denote by $S(t, a)$ the density of individuals of age $a \in [0, a_{max}]$ that are susceptible to the infection at time $t \in [0, T]$. These individuals can become infected with a rate called the force of infection and denoted $\lambda(t, a)$. We assume that a fraction $p$ of these individuals are paucisymptomatic, which means that they will develop very mild to no symptoms, and enter group $I_p$. Note that $p$ is likely to depend on age, but, because it is currently unknown, we assume it to be constant. This class $I_p$ can also be interpreted as the fraction of the population that will not isolate themselves during their infection. Other individuals are assumed to develop more symptomatic infections, either severe $I_s$ with proportion $q(a)$ depending on the age $a$, or mild $I_m$ with proportion $1 - q(a)$.

Each of the three infected host populations are structured in time since infection, so that $I_v(t, a, i)$, $v \in \{p, s, m\}$, denotes the density at time $t$ of individuals of age $a$ that have been infected for a duration $i \in \mathbb{R}_+$. Upon infection, all exposed individuals are assumed to remain non-infectious during an average period $i_{lat}$. Next, they enter an asymptomatic period during which they are infectious. Only $I_m$ and $I_s$ develop significant symptoms after an average time since infection $i_{sympt}$, which can allow them to self-isolate to limit transmission. During their infection, individuals can recover at a rate $h_v(a, i)$ ($v \in \{p, m, s\}$) that depends on the severity of the infection and the time since infection $i$. Severely infected individuals of age $a$ may also die from the infection at rate $\gamma(a, i)$.

The infection life cycle is shown in Fig 1. The total size of the host population of age $a$ at time $t$ is

$$N(t, a) = S(t, a) + R(t, a) + \int_0^\infty (I_p(t, a, i) + I_m(t, a, i) + I_s(t, a, i))\mathrm{d}i. \qquad (1)$$

**Fig 1. The model flow diagram.** Susceptible hosts of age $a$ at time $t$ ($S(t, a)$) are exposed to the virus with a force of infection $\lambda(t, a)$. A fraction $p$ of exposed individuals, which are infected since time $i$, will never develop symptoms and enter the group of paucisymptomatic infections ($I_p(t, a, i)$). The rest will develop symptomatic infections, either severe ($I_s(t, a, i)$) with proportion $q(a)$ depending on age $a$ of individuals, or mild ($I_m(t, a, i)$). Exposed individuals remain non-infectious for a duration $i_{lat}$ after infection. Next, they become asymptomatic infectious and only symptomatic infected will develop symptoms at time $i_{sympt}$ after infection. Infected individuals recover at rate $h_v(a, i)$. Only severely infected of age $a$ die from the infection at rate $\gamma(a, i)$. Notations are shown in Table 1.

**Remark**. *Contrarily to classical SEAIR models, disease-stage progression in our model is not captured by discrete compartments (exposed, asymptomatic, and infected) with exponentially distributed waiting times to switch between compartments. The advantage of our formalism is that disease progression can be modeled using a continuous variable, called the time since infection (in days) denoted here by i. Every infected person then remains in the "infection compartment" from exposure until recovery (or death). Latency from exposed to asymptomatic and time of symptoms onset are not needed for this modeling approach because these are captured through the functions describing the transmission rate, the mortality rate, and the recovery rate at time i post infection. More precisely, the average latency from exposed to asymptomatic ($i_{lat}$) is simply mentioned to define the average time to infectiousness onset ($i_{sympt}$), and also to help the readers to understand the model flow diagram (Fig 1). On the other hand, $i_{sympt}$ is used to define infectiousness reduction factors ($\xi_s$, $\xi_m$) and the mortality rate due to the infection ($\gamma$).*

**Age-structured transmission and severity.** We use two components to model the infection process. First, we define the transmission probability $\beta_v(a, i)$ ($v \in \{p, m, s\}$) for each contact between an infected of age $a$ and a susceptible person, which depends on the time since infection $i$. Second, we introduce the kernel $K(a, a')$ that represents the average number of contacts by unit of time between an individual of age $a'$ and an individual of age $a$. The numerical values of this contact matrix are based on data from an earlier study [43]. The force of infection underwent by susceptible individuals of age $a$ at time $t$ is then given by

$$\lambda(t, a, c) = (1 - c(t, a))$$
$$\times \int_0^{a_{max}} K(a, a') \int_0^\infty (\beta_s(a', i)I_s(t, a', i) + \beta_m(a', i)I_m(t, a', i) + \beta_p(a', i)I_p(t, a', i))\mathrm{d}i \, \mathrm{d}a'. \quad (2)$$

where $c = c(t, a)$ is the percentage of contacts reduction towards people with age $a$, due to control measures, at time $t$. The total force of infection at time $t$ in the whole population is computed as $\int_0^{a_{max}} \lambda(t, a, c)\mathrm{d}a$. The dynamics of newly infected individuals (*i.e.* $i = 0$) in each group is thus defined by

$$\begin{cases} I_s(t, a, 0) &= (1 - p)q(a)\lambda(t, a, c)S(t, a), \\ I_m(t, a, 0) &= (1 - p)(1 - q(a))\lambda(t, a, c)S(t, a), \\ I_p(t, a, 0) &= p\lambda(t, a, c)S(t, a). \end{cases} \quad (3)$$

Further, we assume that only severe infections $I_s$ lead to hospitalization and we denote by

$$H(t) = \int_0^{a_{\max}} \int_{i_{sympt}}^{\infty} I_s(t, a, i)\mathrm{d}i\ \mathrm{d}a \tag{4}$$

the total population hospitalized at time $t$, where $i_{sympt}$ is the average time to symptoms onset. Each individual of age $a$ dies at a rate $\mu(a, H(t))$ at time $t$, defined by

$$\mu(a, H(t)) = \mu_{nat}(a) + \mu_{add}(a, H(t)).$$

where $\mu_{nat}$ denotes the natural mortality rate when hospitals are not saturated. We assume that this rate increases significantly as soon as the number of severe cases exceeds the healthcare capacity $H_{sat}$ and denote by $\mu_{add}$ this additional death rate due to hospital saturation.

We apply the same reasoning by assuming that the disease-related mortality can increase because of hospital saturation. Therefore, severely infected individuals of age $a$ who have been infected since time $i$ die at time $t$ at rate $\gamma(a, i, H(t))$ defined by

$$\gamma(a, i, H(t)) = (\gamma_{dir}(a) + \gamma_{indir}(a, H(t)))\mathbf{1}_{[i_{sympt}, i_{\max}^s]}(i).$$

where $\gamma_{dir}$ and $\gamma_{indir}$ are mortality rates directly and indirectly due to the COVID-19 respectively. The disease-related mortality occurs after the emergence of symptoms and before the mean final time of infection for severe cases, *i.e.* for $i \in [i_{sympt}, i_{\max}^s]$.

Finally, infected individuals of age $a$ infected since time $i$ recover at rates $h_s(a, i)$, $h_m(a, i)$, and $h_p(a, i)$ for severe, mild, and paucisymptomatic infections respectively.

The boundary conditions (3) are coupled with the following equations:

$$
\begin{cases}
\dfrac{\partial S}{\partial t}(t, a) &=& -\mu(a, H(t))S(t, a) - \lambda(t, a, c)S(t, a), \\[2mm]
\left(\dfrac{\partial I_s}{\partial t} + \dfrac{\partial I_s}{\partial i}\right)(t, a, i) &=& -[\mu(a, H(t)) + \gamma(a, i, H(t)) + h_s(a, i)]I_s(t, a, i), \\[2mm]
\left(\dfrac{\partial I_m}{\partial t} + \dfrac{\partial I_m}{\partial i}\right)(t, a, i) &=& -[\mu(a, H(t)) + h_m(a, i)]I_m(t, a, i), \\[2mm]
\left(\dfrac{\partial I_p}{\partial t} + \dfrac{\partial I_p}{\partial i}\right)(t, a, i) &=& -[\mu(a, H(t)) + h_p(a, i)]I_p(t, a, i), \\[2mm]
\dfrac{\partial R}{\partial t}(t, a) &=& \displaystyle\sum_{v \in \{s,m,p\}} \int_0^{\infty} h_v(a, i)I_v(t, a, i)\mathrm{d}i - \mu(a, H(t))R(t, a),
\end{cases} \tag{5}
$$

for any $(t, a, i) \in (0, T] \times [0, a_{\max}] \times \mathbb{R}_+$, with initial conditions (at $t = 0$):

$$S(0, a) = S_0(a), \quad R(0, a) = 0,$$

$$I_s(0, a, i) = I_{s,0}(a, i), \quad I_m(0, a, i) = I_{m,0}(a, i), \quad I_p(0, a, i) = I_{A,0}(a, i),$$

for each $(a, i) \in [0, a_{\max}] \times \mathbb{R}_+$. Numerical values for initial conditions are detailed later. Using equation system (3) and an integration over $i$ of system (5), one may observe that the total population $N$ defined by Eq (1) is strictly decreasing since it satisfies the following inequality:

$$\frac{\partial N}{\partial t}(t, a) \leq -\mu_{nat}(a)N(t, a), \quad \forall a \in [0, a_{\max}], \quad \forall t \geq 0.$$

This is due to the fact that population aging and births are neglected in this model since we consider a time horizon of only one year. Basic properties of the model such as existence and positiveness of solutions is out of the primary scope of our study but these can be specifically

addressed using an integrated semigroup approach and Volterra integral formulation (see *e.g.* [44–47] and references therein). More specifically, one can refer to [31], where the well-posedness of an epidemiological model with a double continuous structure is handled.

## Epidemiological outputs, model parameters and initial conditions

In this section we briefly describe some useful epidemiological outputs, the shape of age-dependent parameters considered for the simulations of model (3)–(5), and the initial conditions. All state variables and other parameters are summarized in Table 1.

**Epidemiological outputs.** In addition to the total number of hospitalized cases $H(t)$ at time $t$ defined by Eq (4), we define additional epidemiological outputs. The first one is the number of non-hospitalized cases ($N_H(t)$)

$$N_H(t) = \int_0^{a_{\max}} \left[ \int_0^{i_{sympt}} I_s(t, a, i) \mathrm{d}i + \int_0^{\infty} (I_m(t, a, i) + I_p(t, a, i)) \mathrm{d}i \right] \mathrm{d}a \qquad (6)$$

which encompasses paucisymptomatic, mildly infected, and severely infected but not yet hospitalized hosts.

For the cumulative number of deaths, we distinguish between those directly due to COVID-19 infections ($D_{dir}^{cum}(t)$), and those indirectly due to the epidemic ($D_{indir}^{cum}(t)$), which originate from the saturation of the health system:

$$D_{dir}^{cum}(t) = \int_0^t D_{dir}(s) \mathrm{d}s, \qquad D_{indir}^{cum}(t) = \int_0^t D_{indir}(s) \mathrm{d}s, \qquad (7)$$

where $D_{dir}(t)$ and $D_{indir}(t)$ are the number of deaths at time $t$ respectively defined by

$$D_{dir}(t) = \int_0^{a_{\max}} \int_{i_{sympt}}^{i_{\max}^s} \gamma_{dir}(a) I_s(t, a, i) \mathrm{d}i \, \mathrm{d}a,$$

$$D_{indir}(t) = \int_0^{a_{\max}} \mu_{add}(a, H(t)) N(t, a) \mathrm{d}a + \int_0^{a_{\max}} \gamma_{indir}(a, H(t)) \int_{i_{sympt}}^{i_{\max}^s} I_s(t, a, i) \mathrm{d}i \, \mathrm{d}a.$$

Every aforementioned output implicitly depends on parameter $c = c(t, a)$, which we will omit in the notations when no confusion is possible. However, for clarity, we do explicitly write this dependence to compare public health measures. The relative performance between two strategies $c_1$ and $c_2$, denoted by $\Delta(c_1, c_2)$, is estimated by assessing the cumulative number of deaths in the whole population during the $T$ days of control period with the strategy $c_1$ relatively to deaths with the strategy $c_2$. Formally, we have

$$\Delta(c_1, c_2) = 1 - \frac{D_{dir}^{cum}(c_1, T) + D_{indir}^{cum}(c_1, T)}{D_{dir}^{cum}(c_2, T) + D_{indir}^{cum}(c_2, T)}.$$

Hence, a relative performance $\Delta(c_1, c_2) = 0.1$ implies that strategy $c_1$ reduces the number of deaths by 10% relatively to strategy $c_2$.

**Model parameters. Mortality rates**. We assume that indirect mortality, *i.e.* not directly due to COVID-19, increases when the number of hospitalisations $H(t)$, at time $t$, exceeds a healthcare capacity threshold $H_{sat}$ (which we approximate with the maximal intensive care capacity). The natural mortality rate then increases by $\mu_{add}(a, H)$ for the whole population, and by $\gamma_{indir}(a, H)$ for severely infected individuals of age $a$. These rates are modeled by logistic

**Table 1. Model variables and parameters.** We show the notations used and indicate references for the numerical values used.

| Param. | Description (unit) | Values [source] |
|---|---|---|
| **State variables** | | |
| $S$ | Susceptible individuals | |
| $I_s$ | Severely infected individuals | |
| $I_m$ | Mildly infected individuals | |
| $I_p$ | Paucisymptomatic infected individuals | |
| $R$ | Recovered individuals | |
| **General parameters** | | |
| $t, T$ | time and final time of simulations (days) | $t \in [0, T]$ (ad hoc) |
| $a, a_{max}$ | age and maximal age of individuals (years) | $a \in [0, a_{max}]$, $a_{max} = 100$ (ad hoc) |
| $i$ | time since infection (days) | $\mathbb{R}_+$ (ad hoc) |
| $i_{lat}$ | average latency from exposed to asympt. (days) | 4.2 [49] |
| $i_{sympt}$ | average time of symptoms onset (days) | $i_{lat} + 1 = 5.2$ [48] |
| $i_{max}^s$ | mean final time of infection for severe cases (days) | $i_{sympt} + 20 = 25.2$ [50] |
| $i_{max}^m$ | mean final time of infection for mild cases (days) | $i_{sympt} + 17 = 22.2$ [50] |
| $\mu_{add}$ | additional death rate (days$^{-1}$) | defined by (8) |
| $\beta_s, \beta_m, \beta_p$ | transmission probabilities (unitless) | computed |
| $\xi_s, \xi_m, \xi_p$ | infectiousness reduction factors (unitless) | defined by (9) and $\xi_p = 0.1$ [22] |
| $h_s, h_m, h_p$ | recovery rates per infection (days$^{-1}$) | defined by (10) |
| $c, c_{max}$ | percentage of contacts reduction and its upper bound | $c \in [0, c_{max}]$, $c_{max} = 0.95$ (assumed) |
| $\gamma_{dir}$ | mortality rate directly due to the COVID-19 (days$^{-1}$) | [48] |
| $\gamma_{indir}$ | mortality rate indirectly due to the COVID-19 (days$^{-1}$) | defined by (8) |
| $p$ | proportion of paucisymptomatic (unitless) | variable |
| $q$ | proportion of symptomatic requiring hospitalisation (unitless) | [48] |
| $B$ | cost of the control measure | variable |

| Param. | Description (unit) | Burkina Faso | France | Vietnam |
|---|---|---|---|---|
| **Specific parameters for each country** | | | | |
| $S_0$ | initial population of susceptible | [51] | [52, 53] | [54] |
| $I_0^{(*)}$ | initial epidemic size | 288 (WHO)[(**)] | 130 [55] | 217 (Ministry of Health) |
| $\mu_{nat}$ | natural death rate (days$^{-1}$) | [56] | [57] | [58] |
| $H_{sat}$ | maximal healthcare capacity (unitless) | 11 [59] | 5000 [55] | 5932 (NIHE)[(**)] |
| $K$ | Contacts matrix of social contacts (days$^{-1}$) | [43] | [43] | [43] |

| Parameters and range for the global sensitivity analysis | |
|---|---|
| Population structure | {Burkina Faso, France, Vietnam} |
| $H_{sat}$ | {10, 100, 500, 2000, 5000, 6000, 50000, 5e+05, 5e+06} |
| $p$ | {0.05 to 0.95} by step of 0.1 |
| $i_{sympt}$ | {1.2 to 9.2} by step of 2 |
| $\xi_p$ | {0.1, 0.3, 0.5, 0.7, 1} |

[(*)]: On Mar, 1st, 2020 in France and on Apr, 1st, 2020 in Burkina Faso and Vietnam.

[(**)]: WHO: World Health Organisation, NIHE: National Institut of Hygiene and Epidemiology.

functions that are arbitrarily chosen as:

$$\mu_{add}(a, H(t)) = \frac{10^{-2}\, \mu_{nat}(a)}{1 + 99\, \exp\left(-10\left(\frac{H(t)}{H_{sat}} - 1\right)\right)},$$

$$\gamma_{indir}(a, H(t)) = \frac{\gamma_{dir}(a)}{1 + 99\, \exp\left(-10\left(\frac{H(t)}{H_{sat}} - 1\right)\right)}. \tag{8}$$

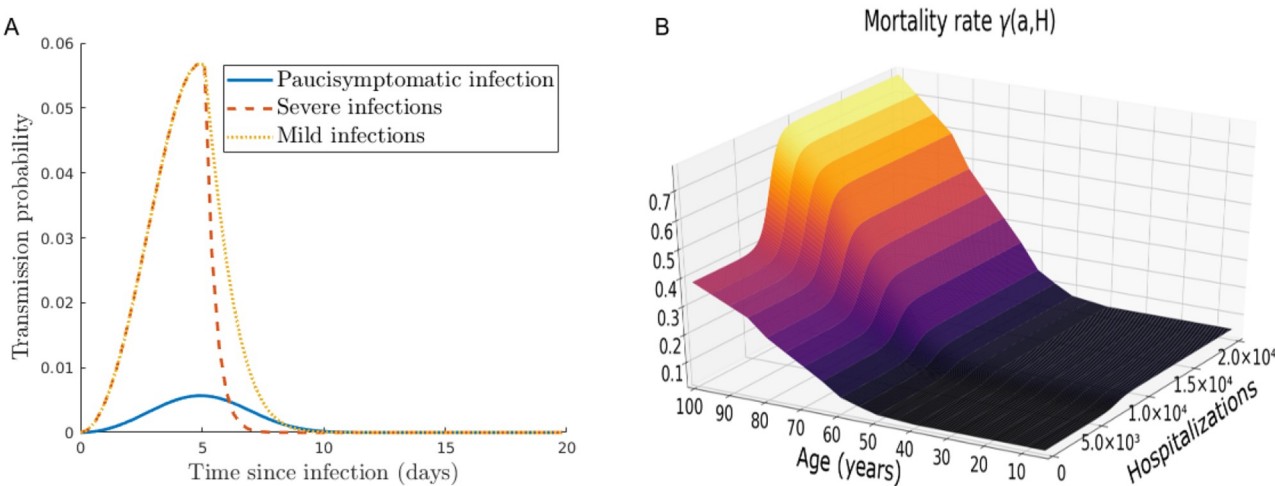

**Fig 2.** (A) Transmission probabilities of paucisymptomatic infections ($\beta_p$), symptomatic severe ($\beta_s$), and mild infections ($\beta_m$). (B) Disease-induced mortality rate with a maximal healthcare capacity $H_{sat} = 5 \times 10^3$.

This choice of functional parameters implies that

$$\mu_{add}(a, 0) \approx 0, \quad \gamma_{indir}(a, 0) \approx 0,$$
$$\mu_{add}(a, H_{sat}) = 10^{-4}\mu_{nat}(a), \quad \gamma_{indir}(a, H_{sat}) = 10^{-2}\gamma_{dir}(a)$$

which means these additional mortalities are negligible when hospitals are not saturated (Fig 2B). In case of full saturation, we have

$$\lim_{H \to \infty} \mu_{add}(a, H) = 10^{-2}\mu_{nat}(a), \qquad \lim_{H \to \infty} \gamma_{indir}(a, H) = \gamma_{dir}(a)$$

for each $a \in [0, a_{\max}]$, meaning that the natural mortality rate is only increased by 1%, while the disease-induced mortality rate $\gamma$ is doubled. Indeed, according to [48], less than 50% of patients in critical care will die in case of no saturation of hospitals.

**Transmission rates**. The infectiousness of an individual aged $a$ who is infected since time $i$, is given by $\beta_v(a, i)$ ($v \in \{s, m, p\}$). Based on estimates described in [22], we assume that $\beta_v$ does not depend on host age $a$, *i.e.*, $\beta_v(a, i) = \beta_v(i)$. As discussed later, this assumption is only made for parameterization purpose and does not impact the general formulation of the model proposed here.

The transmission rate at a given day $i$ post infection of a given type of infectious host is defined such that $\beta_v(i) = \alpha \times \xi_v(i) \times \bar{\beta}(i)$, for $v \in \{s, m, p\}$. As detailed below, $\alpha$ is a scaling parameter obtained from the value of the basic reproduction number $R_0$, which is the mean number of secondary infections caused by an infected host [24]. As [22], we assume that parameter $\bar{\beta}$, which strongly depends on the generation interval, follows a Weibull distribution $\bar{\beta} \sim W(3, 5.65)$. Finally, parameters $\xi_v(i)$ are factors that capture variations in infectiousness based on the type of host. For paucisymptomatic individuals, for instance, these are assumed to be constant ($\xi_p(i) = \xi_p$), while the reduction factor in more symptomatic infections (severe and mild) is assumed to vary after symptom onset to capture admission in a healthcare facility

or self-isolation at home. More precisely, we assume that

$$\xi_s(i) = \begin{cases} 1 & \text{if } i \in [0, i_{sympt}], \\ e^{-\ln(10)(i - i_{sympt})} & \text{if } i > i_{sympt} \end{cases} \quad \text{and}$$

$$\xi_m(i) = \begin{cases} 1 & \text{if } i \in [0, i_{sympt}], \\ e^{-\ln(2)(i - i_{sympt})} & \text{if } i > i_{sympt}. \end{cases} \tag{9}$$

These two functions are chosen arbitrarily by assuming that individuals do not isolate before symptoms onset ($i \leq i_{sympt}$), and that isolation is stronger when symptoms are more severe (Fig 2A). We therefore assume that the transmission probability $\bar{\beta}$ is divided by 10 (respectively 2) every day after the average time of symptoms onset for individuals severely (resp. mildly) infected.

**Recovery rates.** We assume that recovery rates $h_v(a, i)$, $v \in \{s, m, p\}$, of infected individuals of age $a$ infected since time $i$ are independent of $a$ and take the following form:

$$h_s(\cdot, i) = \mathbf{1}_{[i_{\max}^s, \infty]}(i), \quad h_m(\cdot, i) = h_p(\cdot, i) = \mathbf{1}_{[i_{\max}^m, \infty]}(i), \quad \forall i \in \mathbb{R}_+. \tag{10}$$

That is, one can recover from severe (resp. mild and paucisymptomatic) infections only after a time since infection $i_{\max}^s$ (resp. $i_{\max}^m$) corresponding to the mean duration of infection.

**Initial conditions.** The initial susceptible population $S_0$ and epidemic size $I_0$ are given in Table 1. Since, initially, screening is usually restricted to individuals with severe symptoms, we assume that all initial cases are severe infections. Thus, we set $\int_{i_{sympt}}^{i_{\max}^s} \int_0^{a_{\max}} I_{s,0}(a, i) \mathrm{d}a \, \mathrm{d}i = I_0$ as the initial severely infected individuals, which we assume to be uniformly distributed with respect to the time since infection $i$ on the interval $[0, i_{\max}^s]$. Using estimates from [55, 60] on the age distribution of hospitalised people, we derive an estimation of $I_{s,0}(a, i)$ for each $(a, i) \in [0, a_{\max}] \times \mathbb{R}_+$. Next, following the life cycle (Fig 1), we obtain an estimation of the total initial infected population by $\frac{I_{s,0}(a,i)}{(1-p)q(a)}$. From there, we deduce the initial mildly and pauci-symptomatic infected populations, which can be denoted respectively by

$$I_{m,0}(a, i) = \frac{1 - q(a)}{q(a)} I_{s,0}(a, i) \quad \text{and} \quad I_{A,0}(a, i) = \frac{p}{q(a)(1-p)} I_{s,0}(a, i).$$

## Optimal intervention

As explained above, our goal is to find an optimal control strategy that is allowed to vary depending on the number of days since the onset of the epidemic ($t$) and on host age ($a$). In this section, following well established methodology in optimal control theory [13–16, 61], we search for the optimal control effort function $c^*$ that minimizes the objective functional $J : L^\infty(\mathbb{R}_+ \times [0, a_{\max}]) \ni c \mapsto J(c) \in \mathbb{R}$, where

$$J(c) = D_{dir}^{cum}(c, T) + D_{indir}^{cum}(c, T) + \int_0^T \int_0^{a_{\max}} B(a) c^2(t, a) \mathrm{d}a \, \mathrm{d}t,$$

$D_{dir}^{cum}$, $D_{indir}^{cum}$ being the cumulative number of deaths defined by (7), and $B(a)$ the cost associated with the implementation of such control $c$ for the age class $a$. Our aim is to find the function $c^*$

satisfying

$$J(c^*) = \min_{c \in \mathcal{U}} J(c) \tag{11}$$

wherein the set $\mathcal{U}$ is defined by

$$\mathcal{U} = \{c \in L^\infty(\mathbb{R}_+ \times [0, a_{\max}]) : 0 \le c(\cdot, \cdot) \le c_{\max}\},$$

with $c_{\max} \le 1$ a positive constant. That is to say, the function $c^*$ will minimize the cumulative number of deaths during $T$ days, as long as the cost of the control strategy is not too large.

Let $(S, I_s, I_m, I_p, R)$ be a given solution of equation systems (3)–(5) then let $\lambda$ and $H$ be respectively defined by Eqs (2) and (4). After some computations (S1 Text), we find that the adjoint system of (5) can be expressed as

$$
\begin{pmatrix}
\dfrac{\partial z_S}{\partial t}(t, a) \\[2ex]
\dfrac{\partial z_R}{\partial t}(t, a) \\[2ex]
\left(\dfrac{\partial z_{I_s}}{\partial t} + \dfrac{\partial z_{I_s}}{\partial i}\right)(t, a, i) \\[2ex]
\left(\dfrac{\partial z_{I_m}}{\partial t} + \dfrac{\partial z_{I_m}}{\partial i}\right)(t, a, i) \\[2ex]
\left(\dfrac{\partial z_{I_p}}{\partial t} + \dfrac{\partial z_{I_p}}{\partial i}\right)(t, a, i)
\end{pmatrix}
=
$$

$$
\begin{pmatrix}
\mu(a, H(t))z_S(t, a) - \mu_{add}(a, H(t)) \\[2ex]
\mu(a, H(t))z_R(t, a) - \mu_{add}(a, H(t)) \\[2ex]
(\mu(a, H(t)) + h_s(a, i))z_{I_s}(t, a, i) - \mu_{add}(a, H(t)) - \gamma(a, i, H(t))(1 - z_{I_s}(t, a, i)) \\[2ex]
(\mu(a, H(t)) + h_m(a, i))z_{I_m}(t, a, i) - \mu_{add}(a, H(t)) \\[2ex]
(\mu(a, H(t)) + h_p(a, i))z_{I_p}(t, a, i) - \mu_{add}(a, H(t))
\end{pmatrix}
\tag{12}
$$

$$
-
\begin{pmatrix}
\zeta_2(t, a) \int \int K(a, a')(\beta_s(a', i)I_s(t, a', i) + \beta_m(a', i)I_m(t, a', i) + \beta_p(a', i)I_p(t, a', i))\mathrm{d}a'\mathrm{d}i \\[2ex]
0 \\[2ex]
\zeta_1(t, a)\mathbf{1}_{[i_{sympt}, \infty)}(i) + \beta_s(a, i) \int_0^{a_{\max}} \zeta_2(t, a')S(t, a')K(a', a)\mathrm{d}a' + \zeta_3(t, a)h_s(a, i) \\[2ex]
\beta_m(a, i) \int_0^{a_{\max}} \zeta_2(t, a')S(t, a')K(a', a)\mathrm{d}a' + \zeta_3(t, a)h_m(a, i) \\[2ex]
\beta_p(a, i) \int_0^{a_{\max}} \zeta_2(t, a')S(t, a')K(a', a)\mathrm{d}a' + \zeta_3(t, a)h_p(a, i)
\end{pmatrix}
$$

with final conditions $z_S(T, a) = z_R(T, a) = 0$, $z_u(T, a, i) = 0$ and $\lim_{i \to \infty} z_u(t, a, i) = 0$, for any $u$

$\in \{I_s, I_m, I_p\}$ and $(a, i) \in [0, a_{\max}] \times \mathbb{R}_+$, while $\zeta_k$ ($k \in \{1, 2, 3\}$) satisfy the system:

$$
\begin{pmatrix} \zeta_1(t, a) \\ \zeta_2(t, a) \\ \zeta_3(t, a) \end{pmatrix} =
$$

$$
\begin{pmatrix} \frac{\partial \mu}{\partial H}(a, H(t))(S(t, a)(1 - z_s(t, a)) + R(t, a)(1 - z_R(t, a))) \\ [1 - c(t, a)][(1 - p)(q(a)z_{I_s} + (1 - q(a))z_{I_m}) + pz_{I_p}](t, a, 0) - (1 - c(t, a))z_S(t, a) \\ z_R(t, a) \end{pmatrix}
$$

$$
+ \begin{pmatrix} \int_0^\infty \frac{\partial \mu}{\partial H}(a, H(t))(I_s(t, a, i)(1 - z_{I_s}(t, a, i)) + I_m(t, a, i)(1 - z_{I_m}(t, a, i)))\mathrm{d}i \\ 0 \\ 0 \end{pmatrix}
$$

$$
+ \begin{pmatrix} \int_0^\infty \left( \frac{\partial \mu}{\partial H}(a, H(t))I_p(t, a, i)(1 - z_{I_p}(t, a, i) + \frac{\partial \gamma}{\partial H}(a, i, H(t))I_s(t, a, i)(1 - z_{I_s}(t, a, i)) \right) \mathrm{d}i \\ 0 \\ 0 \end{pmatrix}. \tag{13}
$$

Finally, the Hamiltonian $\mathcal{H}$ of (11) is detailed in S1 Text. By solving $\frac{\partial \mathcal{H}}{\partial c} = 0$, it comes that

$$
c^*(t, a) = \max(0, \min(\hat{c}(t, a), 1)), \tag{14}
$$

for every $(t, a) \in [0, T] \times [0, a_{\max}]$, where

$$
\hat{c}(t, a) = \frac{S(t, a)\lambda_0(t, a)[(1 - p)(1 - q(a))z_{I_m}(t, a, 0) + (1 - p)q(a)z_{I_s}(t, a, 0) + pz_{I_p}(t, a, 0)]}{2B(a)},
$$

with $\lambda_0$ detailed in S1 Text.

We also assume that the cost $B(a)$ of the control measure over individuals aged $a \in [0, a_{\max}]$ is proportional to their density in the initial susceptible population $S_0$, i.e.

$$
B(a) = \frac{B^* S_0(a)}{\int_0^{a_{\max}} S_0(u)\mathrm{d}u},
$$

where $B^* \in \mathbb{R}_+$ is a variable parameter characterizing the relative cost in implementing the strategy.

Additionally, one could also factor in the age distribution of the economic cost on the shape of the function $B$. For example, the economic cost is likely to be more important for the working population (i.e. age group 20–60) compared to the older, mostly retired, population. However, in absence of relevant references regarding this topic, we stand with our primary assumption.

The state system (3)–(5) and the adjoint system (12) and (13) together with the control characterization (14) form the optimality system to be solved numerically. Since the state equations have initial conditions and the adjoint equations have final time conditions, we cannot solve the optimality system directly by only sweeping forward in time. Here, we use an iterative algorithm, forward-backward sweep method [8]. In other words, finding $c^*$ numerically first

involves solving the state variables (3)–(5) forward in time, then solving the adjoint variables (12) and (13) backward in time, before finally plugging the solutions for the relevant state and adjoint variables into (14), subject to bounds on the control function. The proof of the existence of such control is standard and is mostly based on Ekeland's variational principle [62]. Therefore, we assume the existence of the solution to the above problem and refer to [13] for additional details.

## Results

Here we consider three countries as case studies: Burkina Faso, France, and Vietnam. Their populations have quite contrasted age-structure and social or physical contacts (Fig 3). In Burkina Faso, the vast majority (96.1%) of the population is less than 60 years old, whereas this is less the case in Vietnam and in France (87.7% and 73.4% respectively) as shown in Fig 3A and 3B. It shows that a higher proportion of the population is older than 60, hence at risk for COVID-19 infection, in France 26.6%, in comparison to Vietnam 12.3%, or to Burkina Faso 3.9%. Also, contacts are more frequent among the older population in France compared to Vietnam (Fig 3D and 3E). By contrast, very few contacts are observed among older populations in Burkina Faso (Fig 3C).

### Global sensitivity analysis

We study the sensitivity of infected individuals, hospitalizations, and deaths to five parameters: the proportion of paucisymptomatic infections ($p$), the average time of symptoms onset ($i_{sympt}$), the infectiousness reduction of paucisymptomatic infections ($\xi_p$), the healthcare capacity ($H_{sat}$), and the population structure (including the natural mortality, the size of the population, age-structure, and social contacts). The variation range of the above parameters is

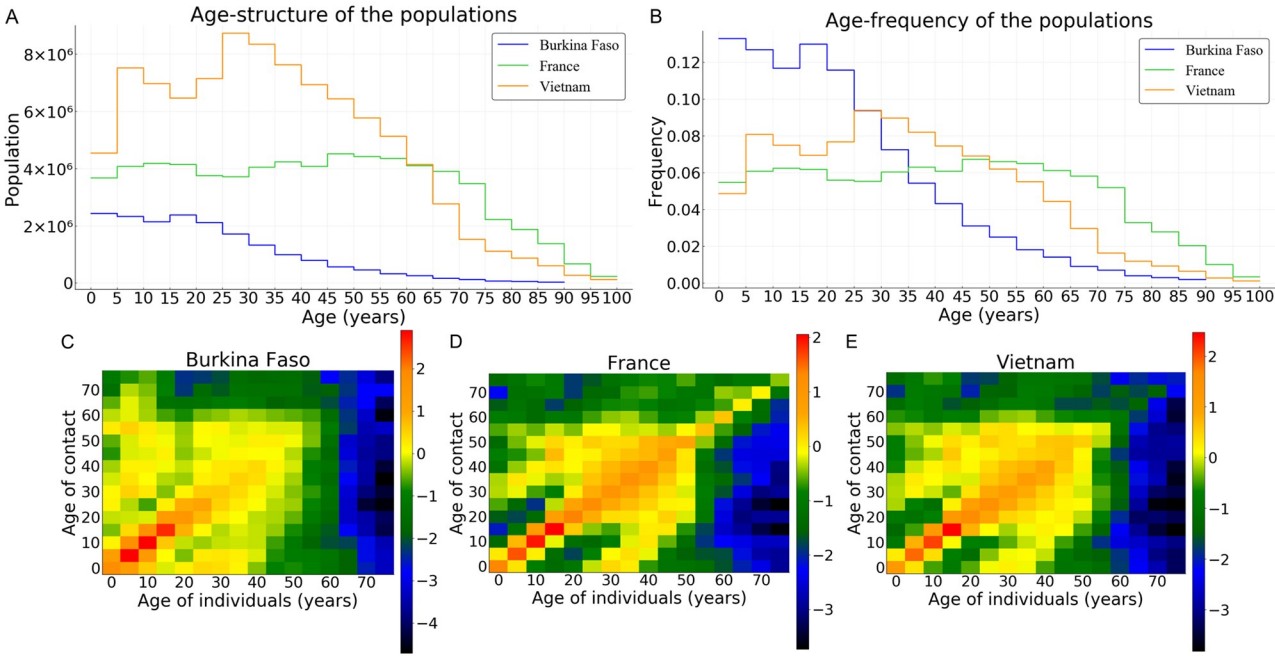

**Fig 3.** The population age-structure of Burkina Faso, France and Vietnam in numbers (A) and frequencies (B). (C)-(E) Contact matrices in the three countries, in log scale where dark color intensities indicate less likely events *i.e.* smaller tendency of having a household member of that age, lower proclivity of making the age-specific contact.

assigned in Table 1. Sensitivity indices are estimated by fitting an ANOVA (Analysis Of Variance) linear model, including third-order interactions, to the data generated by simulation. This ANOVA linear model fitted well with 99% of variance explained.

Overall, the population structure is the main parameter highlighted by the sensitivity analysis. It explains 70% of the variance for the number infected individuals and 40% for hospitalizations and deaths (S1 Fig). The population structure is followed by $\xi_p$, $p$, and $i_{sympt}$ which have a similar impact on the number infected individuals with a slight dominance of $\xi_p$ (S1 Fig). For hospitalizations and deaths, the population structure is followed by $p$ with 40% and 30% of the variance explained respectively; while $\xi_p$ and $i_{sympt}$ have very marginal impact (S1 Fig). Finally, the importance of $H_{sat}$ on the three output variables is largely negligible, with, however, greater importance on deaths as compared to hospitalizations and infections (S1 Fig).

## The basic reproduction number $R_0$

An explicit expression of the $R_0$ of the model defined by Eqs (3)–(5) is difficult to obtain in general. We show in S2 Text that it is possible to write $R_0 = \alpha \times r(\bar{U})$, where $\alpha$ is the scaling parameter introduced earlier, and $r(\bar{U})$ is the spectral radius of the next generation operator $\bar{U}$ defined on $L^1(0, a_{\max})$ by

$$\bar{U} : L^1(0, a_{\max}) \ni v \mapsto S_0(\cdot) \int_0^\infty \int_0^{a_{\max}} K(\cdot, a')\omega(a', i)v(a')\mathrm{d}a' \ \mathrm{d}i \in L^1(0, a_{\max}). \tag{15}$$

where $S_0$ is the initial susceptible population, $K$ is the contact matrix, and $\omega(a, i)$ is the infectiousness of individuals of age $a$ infected since time $i$ (S2 Text). It follows that

$$\alpha = \frac{R_0}{r(\bar{U})}. \tag{16}$$

In our numerical approach, we set $R_0 = 3.3$ [63, 64] for all three countries and corresponding values for $S_0$ and $K$ for each country. We then successively determine $r(\bar{U})$ and $\alpha$ by (15) and (16) respectively.

## Uncontrolled epidemic

We first use the model (3)–(5) to describe the outbreak of the epidemics for all three countries, in absence of public health measure (i.e. $c \equiv 0$), with $R_0 = 3.3$ and other parameters defined previously and summarized in Table 1.

The peak of the epidemics is then reached approximately at day $t = 51$ for hospitalised people, and day $t = 46$ for non-hospitalised people in the France scenario (Fig 4E). Such times to peaks for hospitalised and non-hospitalised people are 47 and 41 (resp. 50 and 45) for Burkina Faso (resp. Vietnam) scenario (Fig 4A, resp. Fig 4I). The delay between the two peaks is explained by the latency time $i_{sympt}$ for symptoms onset (Table 1).

In absence of control measures, the healthcare capacity is quickly exceeded, about twenty days for the 'France' scenario (Fig 4E), and the number of deaths increases sharply from then on. A similar configuration is observed for Vietnam (Fig 4I). By contrast, because of the very low healthcare capacity in Burkina Faso, the health system is exceeded within a few days (Fig 4A). However, this overloading does not have the same consequences in terms of mortality in Burkina Faso compared to France and Vietnam. This is partially explained on the one hand by the fact that less than 4% of the population is above 60 years in Burkina Faso (Fig 3A) and on the other hand by the fact that very few contacts are observed within the older population in Burkina Faso compared to France or Vietnam (Fig 3B and 3D).

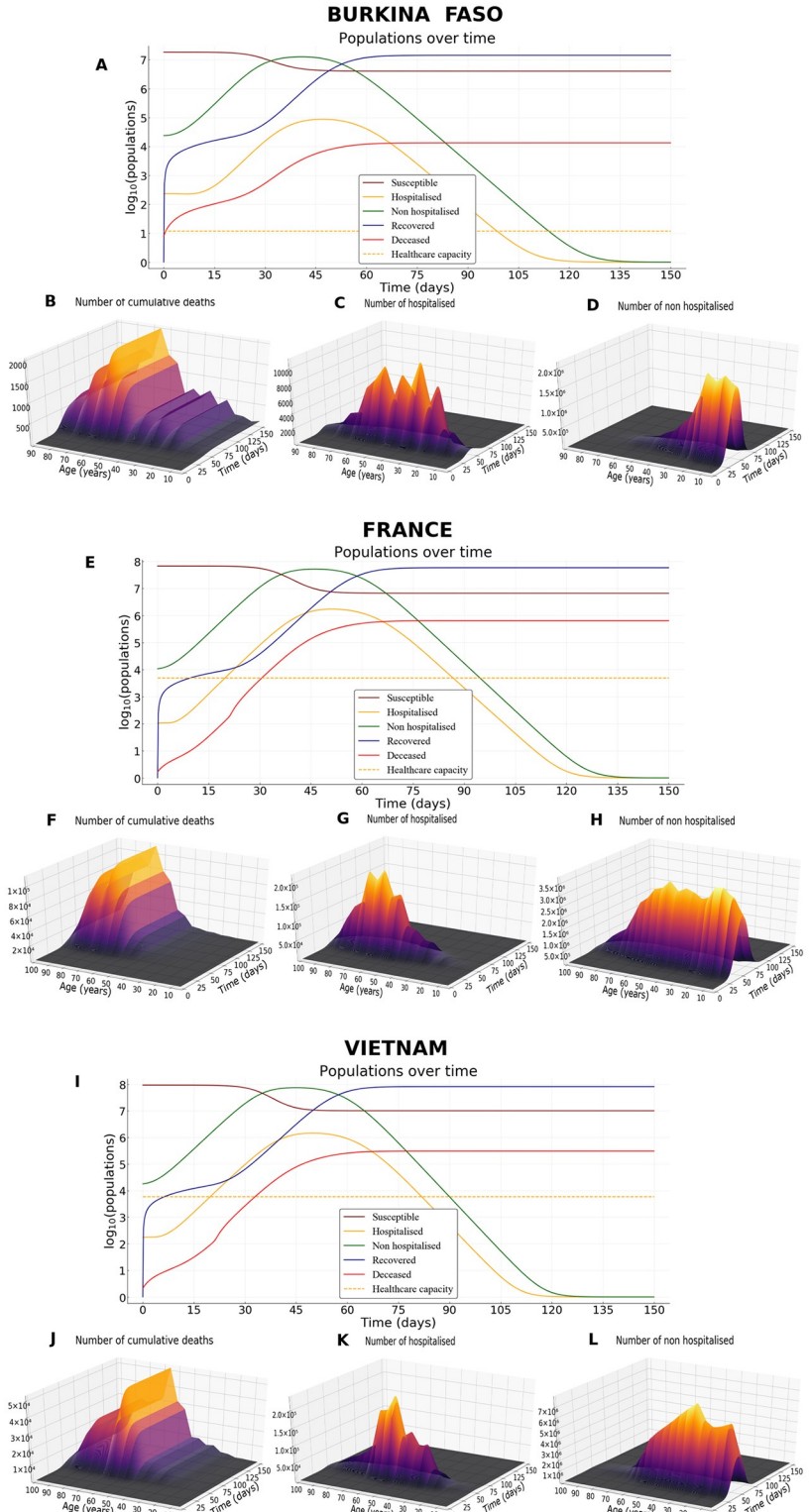

**Fig 4. Epidemic scenario without any control measures.** (A) Dynamics of epidemiological outputs, (B) number of cumulative deaths, (C) number of hospitalised and (D) non-hospitalised people in Burkina Faso. (E-H) As for (A-D) but in France. (I-L) As for (A-D) but in Vietnam. Parameter values are default in Table 1, $R_0$ = 3.3, and the proportion of paucisymptomatic infections is $p$ = 0.5.

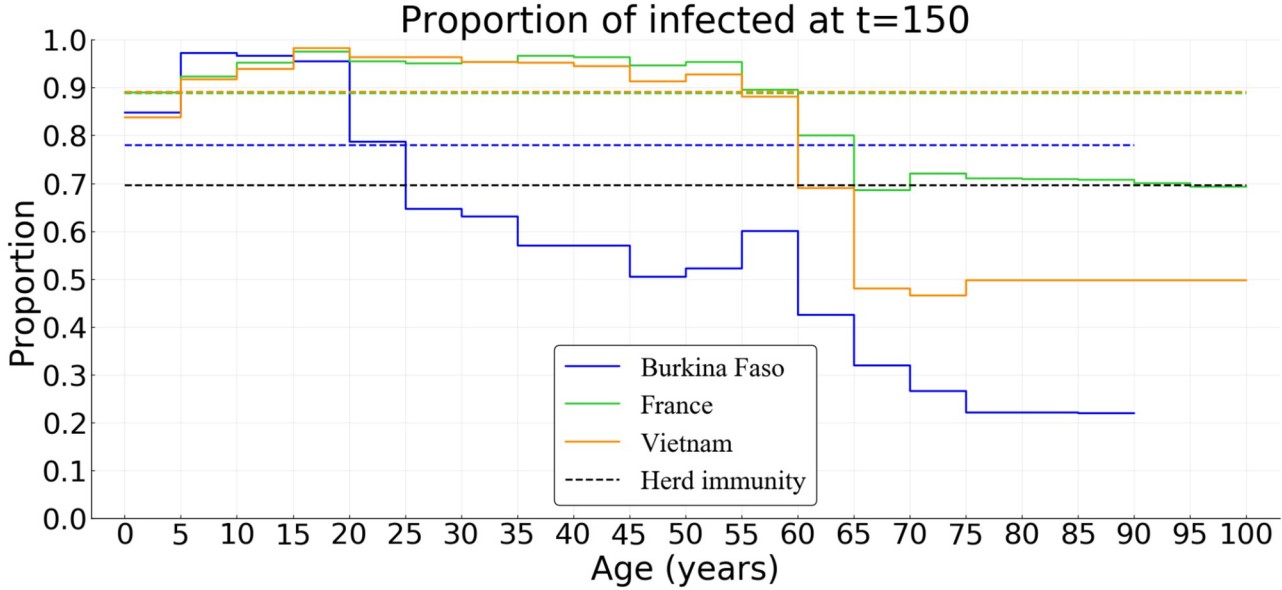

**Fig 5. Simulated age distribution of the proportion of the population infected after 150 days in Burkina Faso, France, and Vietnam in absence of control measures.** Parameter values are default (Table 1, $R_0 = 3.3$, and the proportion of paucisymptomatic infections is $p = 0.5$.

At the end of the simulation ($t = 150$ days), without any control measures, the herd immunity threshold ($1 - 1/R_0 \approx 69.7\%$) is clearly reached in France and in Vietnam (Fig 5), where the average size of the epidemic (severe, mild, and paucisymptomatic infections) is close to 90%. The threshold is also reached in Burkina Faso but only with an epidemic size of 78% (Fig 5). In all three countries, the proportion of the population less than 20 years old that has been infected is around 93%. In the [20–60] group, we find a similar percentage in France and Vietnam (94%), but only 65% in Burkina Faso. This proportion then decreases for the population older than 60, more or less quickly depending on the country: it is around 73% in France, 56% in Vietnam, and 33% in Burkina Faso. Finally, among the infected population, more than 98% are less than 60 in Burkina Faso, while this proportion is 92% in Vietnam and 76% in France. This age structure of infected populations is particularly important since most of the infections that occur in the young population do not require hospitalization (Fig 4C, 4G and 4K) while people older than 60 represent the age class with the highest cumulative number of deaths (Fig 4B, 4F and 4J).

## Optimal intervention

We now investigate the result of implementing an optimal intervention that accounts for the age structure of the population. Strategies performances are here compared in terms of the cumulative number of deaths for three costs of control measures; low ($B^* = 10^2$), intermediate ($B^* = 10^3$), and high ($B^* = 10^4$).

The optimal control strategy varies in time and depends on host age. In general, regardless of the country (Burkina Faso, France, or Vietnam), the control is stronger early in the epidemic and for older populations (Fig 6, S2 and S3 Figs). Overall, the level of optimal control is lower in Burkina Faso compared to France and Vietnam (Fig 6, S2 and S3 Figs). If the cost of implementing the measures $B^*$ is intermediate or high, the optimal control is almost restricted to individuals above 55 and to the first third of the time interval considered, with a significant reduction in deaths (Fig 6D and 6E, S2 and S3 Figs). In France, the relative performance of the

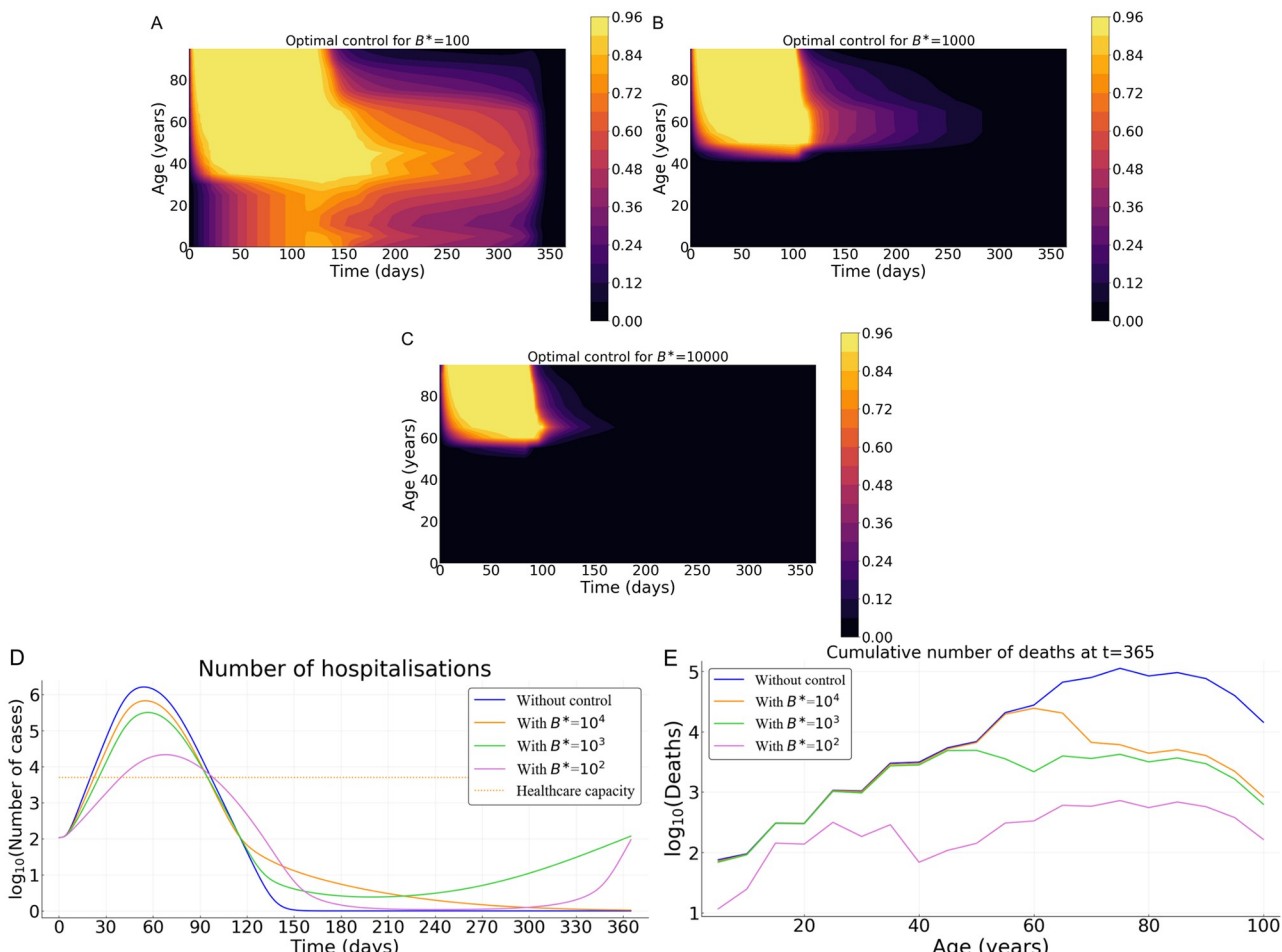

**Fig 6. Optimal control strategy ($c^*$) as a function of the cost of the control measures in France.** Intensity of the control as a function of time and host for for (A) relatively low $B^* = 10^2$, (B) an intermediate $B^* = 10^3$, and (C) a high $B^* = 10^4$ cost. (D) Prevalence of hospitalized patients as a function of the strategy and the cost. (E) Cumulative deaths per age at the end of the time interval (when $T = 365$ days). Parameter values not related to the control are identical to Fig 4. Cases of Burkina Faso and Vietnam are shown in S2 and S3 Figs.

optimal control $c^*$ compared to a 'doing nothing' scenario ($\Delta(c^*, 0)$) is at least 92% (resp. 82%) when the cost is $B^* = 10^3$ (resp. $10^4$). For Burkina Faso, $\Delta(c^*, 0)$ is at least 50% (resp. 4%) when $B^* = 10^3$ (resp. $10^4$). Finally, for Vietnam $\Delta(c^*, 0)$ is at least 87% (resp. 62%) when $B^* = 10^3$ (resp. $10^4$). In the case of Burkina Faso, note that the level of the optimal control is quite low when the cost of implementation is high (S2 Fig), and, as a result, the effect of this control in reducing mortality at the population level is negligible. This is due to the relatively small number of deaths in the whole population in Burkina Faso without any control measures (Fig 4A).

If the implementation of the control measure comes at a low cost ($B^* = 10^2$), the optimal control significantly extends to younger populations in all three countries (Fig 6A, S2 and S3 Figs), with a maximum intensity reached near the 4th month of the epidemics and a steady decrease until the end of the control period. Overall, the optimal control lasts less longer in Burkina Faso (S2 Fig) compared to the cases of France and Vietnam (Fig 6A and S3 Fig). At first, the control is mainly applied to people above 35 in all three countries (Fig 6A, S2 and S3 Figs). But, while the control extends to people less than 35 in France and Vietnam after 2 or 3 months (Fig 6A and S3 Fig), such an extension is very moderate (or even negligible) in Burkina

Faso (S2 Fig). The resulting reduction in the number of deaths is very pronounced with a relative performance $\Delta(c^*, 0)$ of at least 80% (resp. 99%, 97%) in Burkina Faso (resp. France, Vietnam).

## Performance and practical implementation

To illustrate how the strategy identified using optimal control theory outperforms constant uniform control exerted over the whole population or its younger fraction, we derive optimal strategies that do not vary in time and use the same amount of 'resources' (that is the same cumulative cost). Assuming a relatively high cost $B^* = 10^3$, we first investigate a control strategy that targets the younger fraction of the population (Fig 7A), a second strategy that uniformly targets the whole population (Fig 7B). Both strategies have a control level $c_{max} = 0.95$ and vary in duration (the total amount of resource used being constant).

In France, when targeting the population uniformly, the epidemic is under control for approximately 60 days. However, once the control resources are exhausted, the epidemic reemerges (Fig 7C). With the (longer) control over the younger fraction of the population, the first epidemic peak is slightly delayed and the epidemic appears to be under control for a

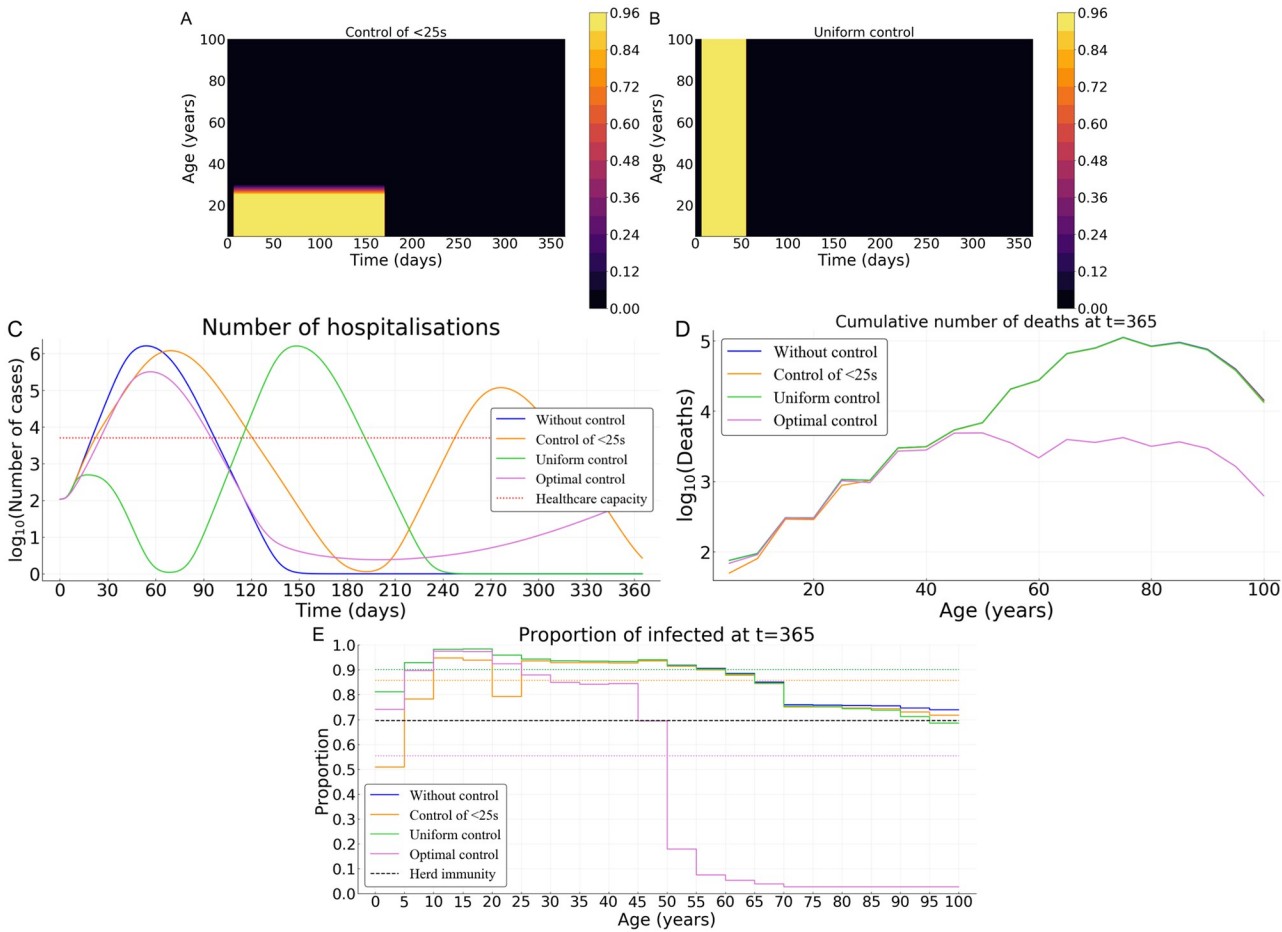

**Fig 7. Comparing optimal control with uniform control of the whole population of over its younger fraction in France.** (A) Illustration of the control over the young population and (B) uniform control of the whole population. (C) Number of hospitalizations. (D) Cumulative deaths per age at final time $T = 365$ days. (E) Age distribution of the proportions of the population that have been infected before one year. Here, we assume $B^* = 10^3$ and $p = 0.5$.

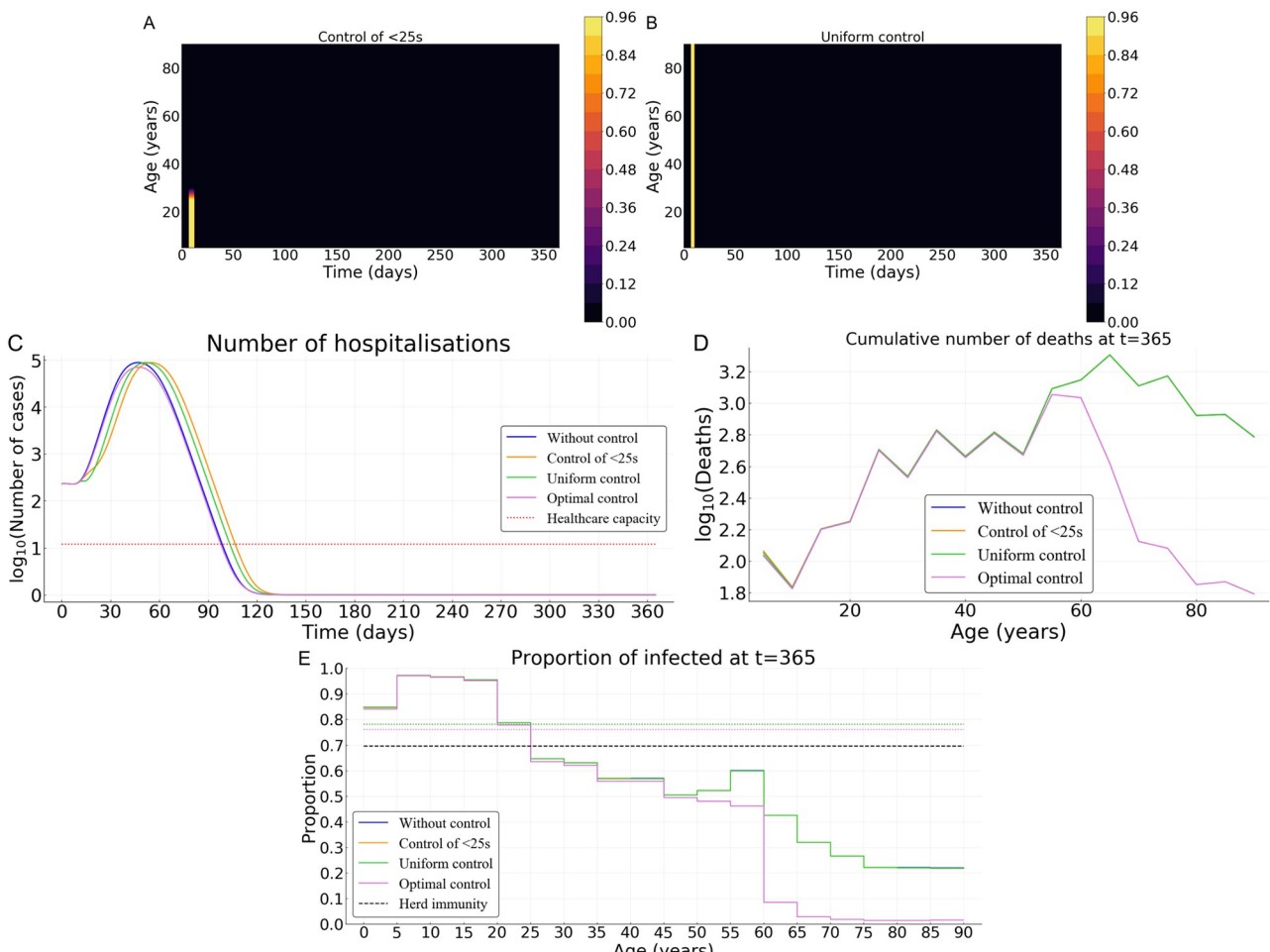

**Fig 8. Comparing optimal control with uniform control of the whole population or over its younger fraction in Burkina Faso.** (A) Illustration of the control over the young population and (B) uniform control of the whole population. (C) Number of hospitalizations. (D) Cumulative deaths per age at final time $T = 365$ days. (E) Age distribution of the proportions of the population that have been infected before one year. Here, we assume $B^* = 10^3$ and $p = 0.5$.

longer time (180 days). Unfortunately, resources also become exhausted and a second peak appears a few months later (Fig 7C). In both cases, *i.e.* targeting the whole population or its younger fraction only, the results are similar. A uniform control leads to a cumulative mortality over the time period of interest comparable to that without any control measure (Fig 7D). The performance of the optimal control relative to a uniform control is approximately 92%. In the end, only 55% of the whole population has been infected with the optimal control but at least 85% with a uniform control (Fig 7E). Similar results are obtained for the case of Vietnam (S4 Fig).

By contrast, for the case of Burkina Faso, regardless of the control strategy (optimal, uniform without targeting, and uniform targeting the younger fraction) the proportion of the population that is infected is approximately the same as without control (78%). The herd immunity threshold ($1 - 1/R_0 \approx 69.7\%$) is then reached for all the three control measures (Fig 8E) and the epidemic cannot restart (Fig 8C). The cumulative mortality with a uniform control exerted over the whole population or over its younger fraction is comparable to that without any control measure (Fig 8D). However, the optimal control performs at least 50% better than

the uniform control (whether it concerns the whole population or its younger fraction). This result holds despite the nearly identical proportions of infections in the population under 60 years of age (Fig 8E).

A practical issue regarding the implementation of the optimal control strategy we identified is that it is a continuous function. One possibility to address this problem is to use step functions. In Supplementary S5 Fig, we subdivided the population into 10-year amplitude classes and updated the control every 3-weeks. Importantly, even though it is assumed to be constant during each 3-weeks period for each age-class, the control intensity directly originates from the results of the continuous optimal control strategy. This discrete implementation of the optimal strategy achieves similar efficiencies (S5 Fig), with a relative performance of 91% compared to a doing nothing scenario.

## Discussion

Non-pharmaceutical public health interventions can be implemented either to mitigate the COVID-19 epidemic wave or to suppress the wave long enough to develop and implement a vaccine or a treatment. Here, we explicitly factor in the age heterogeneity of the host population in the identification of the optimal allocation of the control efforts. We focus on three countries (Burkina Faso, France, Vietnam) with contrasted population age-structures and social or physical contacts (Fig 3).

We use optimal control theory [8] to characterize an optimal strategy that significantly reduces the number of deaths while being sustainable at the population level. Our formulation assumes a quadratic cost for the control effort. Overall, the optimal control lasts less in Burkina Faso compared to France and Vietnam. With this strategy, we find that the intensity of the control is always relatively high on the older fraction of the population during at least a hundred days, before decreasing more or less rapidly depending on the cost associated with the control and the social structure of the host population. The control over the younger fraction of the population is weak and only occurs when the cost associated with the optimal control is relatively low. However, while the control applies to the younger population in France and Vietnam after 2 or 3 months, this is very moderate (or even negligible) in Burkina Faso. This late control over the younger part of the population mimics the results found in [10], which did not include host age but found that control did not peak right away. Intuitively, if control strategies come at a high cost for the population, it is best to focus on the age classes that are the most at risk. Conversely, if the control measures are more acceptable to the population, the optimal strategy is to aim wide to completely suppress the epidemic wave.

Information on the natural history of paucisymptomatic infections of COVID-19 remains relatively poorly documented [65, 66]. We estimated that a proportion $p$ of infected individuals remain asymptomatic throughout their infection, but this proportion remains largely unspecified in the literature [65, 66]. We performed a sensitivity analysis of $p$ on the optimal control strategy $c^*$. Overall, the proportion of paucisymptomatic infections has marginal effects on the optimal control strategy (S6 Fig). The optimal control remains strong over the older population from the beginning of the epidemic, before being progressively relaxed. The control over the younger population is weaker and occurs only if the control cost itself is low. But, the level of control over the younger fraction increases when the proportion of paucisymptomatic infections decreases. Further, for high values of $B^*$ ($10^3$ or $10^4$), the shape of the optimal control is qualitatively the same when the proportion $p$ varies, except for extremely high values of $p = 0.9$ and $B^* = 10^4$, for which the control becomes low for the whole population (S6 Fig). The interpretation is that if the epidemic cannot be stopped, the best strategy is then to reduce mortality by protecting the population the most at risk (here the older population). However, with a low

value of $B^*$ ($10^2$), different shapes of the optimal control yield the same result since there are enough resources to stop the epidemics.

Given the leverage represented by school and university closures, we investigated the effect of targeted control measures over individuals aged under 25. Our results show that NPIs targeting the younger fraction of the population are not very efficient in reducing cumulative mortality unless they can be implemented strongly and for a relatively long period. Indeed, the number of deaths recorded when implementing a control only over the younger population is similar to a doing nothing scenario for cases of Burkina Faso, France, and Vietnam (Figs 7 and 8, and S4 Fig). This result seems independent from the age-structure and social contacts of the population considered. However, as we further discuss below, the variation of the transmission probability with age could matter.

The formulation of the objective functional considered here aims to minimize the cumulative number of deaths. However, other objective functions could be envisaged and factor in, for example, long-term hospitalizations or long-term health consequences. Practically, this could be implemented similarly to the one presented here by considering the number of hospitalizations as the variable to be minimized and costs associated with long-term hospitalizations and long-term health consequences. This formulation may indeed be interesting to investigate in detail but would deserve a dedicated study.

Our model is an extension of models based on ordinary differential equations that tackled the issue of the optimal control of COVID-19 outbreaks [9–12]. However, the whole population is here structured by age ($a$) and additionally by the time since infection ($i$) for infectious individuals, which echoes the model developed in [64] using a discrete-time formulation of the infection. With our continuous structure, we show that the number of new cases $I_N(t, a)$ at time $t$ in individuals of age $a$ is given by the renewal equation

$$I_N(t, a) = S_0(a) \int_0^\infty \int_0^{a_{\max}} K(a, a')\omega(a', i)I_N(t - i, a')\mathrm{d}a' \ \mathrm{d}i,$$

where $K$ is the contact matrix and $\omega(a, i)$ is the infectiousness of individuals aged $a$ which are infected since time $i$ (S2 Text). For parameterization purposes, we assume that $\omega(a, i)$ is the product between the proportion of individuals of age $a$ in the whole population and the infectiousness $\bar{\beta}(i)$ of individuals infected since time $i$. This is potentially a limitation —not in the model formulation proposed here, but rather with regards to our parameterization using the existing literature— since infectiousness $\bar{\beta}$ could depend on the age $a$, thereby creating an additional heterogeneity to that of the time since infection $i$. This issue can be particularly important since some studies suggest a low risk of transmission in the young population (*e.g.* [67]). On the other hand, although superspreading events (of young people) have been documented, there is still much uncertainty about their relative role in the spread of the epidemic and about their origin (superspreading could be linked to environmental conditions, such as massive gatherings, rather than individual properties). Therefore, assuming independence from age appears to be the most parsimonious assumption given the current data.

Another potential limitation is the lack of gender structure and comorbidities in the model formulation. Given the observed male-biased in mortality during the COVID-19 pandemic, it has been suggested that males are more at risk of developing severe infections [68]. This heterogeneity could readily be introduced in the model if its biological importance is further demonstrated.

Contact networks have an important role in transmission dynamic models [69]. Epidemic models that determine which interventions can best control an outbreak may benefit from accounting for social structure and mixing patterns. Contacts are known to be assortative with

age across a given country, but regional differences in age-specific contact patterns are noticeable [43]. The current model could be modified to explore epidemiological dynamics in a spatially structured population with non-homogeneous mixing, *e.g.* by using a meta-population model [70].

Another potential extension of the model would be to allow for the isolation of symptomatic cases and their contacts, following the method developed in [71] and applied recently to digital contact tracing [22]. Indeed, these measures strongly depend on the relative timing of infectiousness and the appearance of symptoms, which are both well-captured by the formulation of our model. However, this also raises technical challenges due to the double continuous structure. Being able to identify age classes to follow in priority with contact tracing could be, though, an asset in controlling epidemic spread.

Finally, a challenge of optimal control theory remains its application in the field. We show in our results that technical obstacles related to the continuous nature of the strategy identified can readily be addressed by using discrete time periods. However, as mentioned above, the choice of the objective function remains delicate because its social implications. Being able to factor in multiple feed-backs (from clinicians, economists, but also the general public) could make optimal control approaches even more impactful in epidemiology.

## Supporting information

**S1 Fig. Global sensitivity analysis.**
(TIF)

**S2 Fig. Optimal control strategy in Burkina Faso.**
(TIF)

**S3 Fig. Optimal control strategy in Vietnam.**
(TIF)

**S4 Fig. Comparing optimal control with uniform control of the whole population or over its younger fraction in Vietnam.**
(TIF)

**S5 Fig. Practicability of the age-structured optimal control.**
(TIF)

**S6 Fig. The effect of paucisymptomatic infections, through their proportion $p$, on the optimal control.**
(TIF)

**S1 Text. Computations of the adjoint system.**
(PDF)

**S2 Text. The basic reproduction number.**
(PDF)

## Acknowledgments

We thank the entire ETE modeling team composed of Thomas Bénéteau, Gonché Danesh, Baptiste Elie, Yannis Michalakis, Bastien Reyné, Christian Selinger for discussion.

## Author Contributions

**Conceptualization:** Quentin Richard, Samuel Alizon, Ramsès Djidjou-Demasse.

**Formal analysis:** Quentin Richard, Ramsès Djidjou-Demasse.

**Funding acquisition:** Samuel Alizon, Marc Choisy, Mircea T. Sofonea, Ramsès Djidjou-Demasse.

**Investigation:** Quentin Richard, Ramsès Djidjou-Demasse.

**Methodology:** Quentin Richard, Ramsès Djidjou-Demasse.

**Supervision:** Ramsès Djidjou-Demasse.

**Writing – original draft:** Quentin Richard, Samuel Alizon, Ramsès Djidjou-Demasse.

**Writing – review & editing:** Quentin Richard, Samuel Alizon, Marc Choisy, Mircea T. Sofonea, Ramsès Djidjou-Demasse.

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
