## [Decision Letter · Decision Letter 0]

14 Jan 2021

Dear Dr. DJIDJOU-DEMASSE,

Thank you very much for submitting your manuscript "Age-structured non-pharmaceutical interventions for optimal control of COVID-19 epidemic" for consideration at PLOS Computational Biology. As with all papers reviewed by the journal, your manuscript was reviewed by members of the editorial board and by several independent reviewers. The reviewers appreciated the attention to an important topic. Based on the reviews, we are likely to accept this manuscript for publication, providing that you modify the manuscript according to the minor review recommendations by Reviewer 1 as noted below.

Sincerely,

Gabor Balazsi

Guest Editor

PLOS Computational Biology

Tom Britton

Deputy Editor

PLOS Computational Biology

[LINK]

Reviewer's Responses to Questions

**Comments to the Authors:**

Reviewer #1: The authors satisfactorily addressed my concerns.

However, my perception is that the authors did not sufficiently dissect (and slightly oversimplified) their results, which might be better interpreted. My comments below reflect this perception.

* Author summary.

I do not have exactly the same interpretation as the authors regarding France and Vietnam, for which the optimal strategy seems to be composed of two different periods:

1/ The first part of the optimal strategy consists in protecting the most vulnerable people (60+ years old), and if the cost is intermediate, to protect 40/50+ people as well (Fig. 6 and S3), which helps controlling the epidemic (temporarily). Indeed the epidemic restarts at the end of the control period if the cost is low or intermediate, while it does not restart if the cost is high. In other words, if the cost is high, the only thing one can do is to protect older people during the uncontrolled epidemic, which lasts a hundred days.

2/ The second part of the optimal strategy targets epidemic control rather than the protection of older people. After an initial period of a hundred days, the number of cases is below the health care capacity, and the optimal control applies to age classes within the interval [40,60] if the cost is low or intermediate. Strikingly, the optimal control does not target 60+ people any longer. This is particularly clear in Vietnam (Fig. S3). The same trend applies to France (Fig. 6). In particular, the optimal control is zero for the 80+ age-classes.

The optimal strategy in Burkina Faso is composed of only the first part: protect older people (what “older” means depends on the cost), and do not try to control the epidemic.

Consequently, I would suggest rephrasing the abstract or writing ([] means removal):

“This strategy consists in rapidly intervening in older populations [to protect the older people during the initial phase of the epidemic and (if the cost is intermediate or low) to control the epidemic], before progressively alleviating this control. Interventions in the younger population can occur later if the cost associated with the intervention is low. This intervention [targeted at younger people] aims at suppressing the epidemic instead of [] protecting the [older people].”

* Introduction – paragraph starting line 11

Optimal control (OC) is presented in a too restrictive (or simplified) way for journal as technical as PLoS Comp. Biol. OC is not restricted to Pontryagin’s Maximum Principle and “open-loop” control (meaning that the control variable depends only on time) with a finite time horizon. OC can also accommodate infinite horizon and state-feedback control, using Hamilton-Jacobi-Bellman equations for instance.

Consequently, I would suggest rephrasing the introduction or writing ([] means removal):

“Things become [] more challenging when the intervention parameter value is a function of time. Optimal control theory [8], [] specifically addresses this issue by identifying a function of time such that [] some criterion is optimised. This has allowed studies to identify optimal non-pharmaceutical interventions to control infectious diseases such as influenza and COVID-19 [9-12].” […]

"Accounting for two dimensions, time and host age, make the optimisation procedure [much] more challenging because [optimal control theory is usually] applied to ordinary differential equations (ODEs) -something very common- while here we are working on partial differential equations (PDEs) -which is less common, and [much] more challenging.

* Line 68: please place here the sentence from current line 106.

“Note that p likely depends on age, but because is it totally unknown, we assume it is a constant.”

* Remove “The time since infection grows linearly with time, according to the derivative with respect to i.”

* Fig. 7’s legend: “France” should be replaced with “Vietnam”.

* Line 352: unclear what “despite their same proportion of infected individuals” means, as the lines cannot be distinguished in Fig. 8(e). Same concern with the last part of the sentence re. Fig. 8(d). This echoes a general comment: please make sure that every color can be seen on the graphs.

* Line 375: remove “shorter”.

* Line 381: replace “extends” with “applies” - see my earlier comments relative to the Author summary. By the way, if my interpretation makes sense, I would suggest updating that part of the Discussion as well.

* Fig. S6: why did you consider the proportion of paucisymptomatic infections p in [0, 0.5] while it was in [0, 0.95] in the sensitivity analysis (Table 1)? I would find it more convincing to show the optimal strategies for larger values of p.

Reviewer #2: The authors have addressed most of my major claims, and so I accept the manuscript.

**Have all data underlying the figures and results presented in the manuscript been provided?**

Reviewer #1: None

Reviewer #2: None

PLOS authors have the option to publish the peer review history of their article (what does this mean?). If published, this will include your full peer review and any attached files.

Reviewer #1: No

Reviewer #2: No
---

## [Editor Report · Decision Letter 1]

7 Feb 2021

Dear Dr. DJIDJOU-DEMASSE,

We are pleased to inform you that your manuscript 'Age-structured non-pharmaceutical interventions for optimal control of COVID-19 epidemic' has been provisionally accepted for publication in PLOS Computational Biology.

Best regards,

Gabor Balazsi

Guest Editor

PLOS Computational Biology

Tom Britton

Deputy Editor

PLOS Computational Biology

The authors have satisfactorily addressed all of the remaining comments.

---

## [Editor Report · Acceptance letter]

16 Feb 2021

PCOMPBIOL-D-20-02129R1 

Age-structured non-pharmaceutical interventions for optimal control of COVID-19 epidemic

Dear Dr Djidjou-Demasse,

I am pleased to inform you that your manuscript has been formally accepted for publication in PLOS Computational Biology. Your manuscript is now with our production department and you will be notified of the publication date in due course.

With kind regards,

Alice Ellingham
